# Late Chronotype is Associated with Adolescent Asthma: Assessment Using the Korean-Version MCTQ

**DOI:** 10.3390/ijerph17093000

**Published:** 2020-04-26

**Authors:** Chang Hoon Han, Jaeho Chung

**Affiliations:** 1Department of Internal Medicine, National Health Insurance Service Ilsan Hospital, Goyang 10444, Korea; 2Department of Internal Medicine, International St. Mary’s Hospital, Catholic Kwandong University College of Medicine, Incheon 22711, Korea

**Keywords:** sleep, chronotype, asthma, adolescents

## Abstract

*Objectives:* In the study, we explored whether sleep chronotypes are associated with asthma in adolescents. *Methods:* We analyzed 24,655 physician-diagnosed adolescent asthmatic patients and 253,775 non-asthmatic adolescent patients from the Korea Youth Risk Behavior Web-based Survey (KYRBWS). Socioeconomic factors, health behaviors factors, psychological factors, and sleep parameters were assessed using the Munich Chronotype Questionnaire (MCTQ). Logistic regression after adjusting for multiple confounders was used to explore the association between sleep chronotype and asthma. *Results:* The asthmatic adolescent group slept less (≤5 h: 24.3% vs. 23.2%) than the non-asthmatic adolescent group. Mean sleep duration (430.6 ± 95.6 vs. 433.5 ± 93.6 min), midpoint of sleep on school-free days (MSF; 255.9 ± 75.9 vs. 258.3 ± 73.6 min), midpoint of sleep on school days (MSW; 199.1 ± 49.1 vs. 200.1 ± 48.4 min), sleep duration on school days (SDW; 398.2 ± 98.1 vs. 400.2 ± 96.8 min), and sleep duration on school-free days (SDF; 511.8 ± 151.9 vs. 516.7 ± 147.2 min) were significantly lower, sleep satisfaction was significantly poorer (low sleep satisfaction: 41.3% vs. 37.5%), and late chronotype was significantly higher in the asthmatic adolescent (21.1% vs. 20.0%). After adjusting for multiple confounders, late chronotype was significantly associated with an increased frequency of adolescent asthma (OR 1.05; 95% CI 1.01–1.09) compared to intermediate chronotypes. *Conclusions:* Although our study shows a very modest association (OR of 1.05 in the fully adjusted model), we show that the late sleep chronotype is associated with asthma in adolescents in South Korea.

## 1. Introduction

Individuals differ in terms of the times at which they go to sleep and wake up, and can be classified based on their diurnal type (chronotype) as “early birds” (morning types) or “night owls” (evening types). Individuals with sleep patterns between the two types are classified as intermediate. The chronotype reflects individual sleep time preference. Chronotypes can be evaluated in several ways, including using the Morning–Eveningness Questionnaire (MEQ) and the Munich Chronotype Questionnaire (MCTQ); the Korean versions of these instruments have been validated [1]. Chronotype is a biologically determined preference but can be modified [2]. Previous reports have shown that chronotype is associated with sex [3], age [4], education level [5], area of residence (urban or rural) [6], physical activity [7], smoking [8], and alcohol use [9]. In addition, previous reports have demonstrated that certain chronotypes, particularly the evening type, are associated with depression [10], fibromyalgia [11], poor sleep quality [5], unhealthy dietary habits [9], cancer [12], overeating [13], and type 2 diabetes and hypertension [14].

Asthma symptoms are often exacerbated by night-time physiobiological changes (e.g., airway inflammation, accidental coughing, wheezing, and breathing difficulties). Eventually, asthma may reduce sleep duration and the extent of Stage 4 sleep and trigger awakening during sleep [15,16]. Asthma may adversely affect sleep biology, but few reports have explored whether sleep chronotype might affect asthma. Although chronotypes have been investigated in a general adult asthmatic population, adolescent asthmatics have not been evaluated. Thus, we explore the sleep chronotype of asthmatic and non-asthmatic adolescent patients and analyze whether there is an association between sleep chronotype and adolescent asthma based on nationally representative, school-based self-reported data on Korean adolescents.

## 2. Methods

### 2.1. Study Participants

This school-based national-wide study was performed by the Korea Center for Disease Control and Prevention and was approved by the Institutional Review Board of the Korea Center for Disease Control and Prevention. Data were collected by the Korea Youth Risk Behavior Web-based Survey (KYRBWS) from 2013 to 2017 from adolescents aged 12 to 18 years. All adolescents were stratified by 43 regional and school-related variables, and samples were defined based on school and class. The online survey was an anonymous self-administered questionnaire containing 134 questions on 14 health-related behaviors of Korean adolescents; the response rate was 95.5% [17]. After the survey procedures were completely explained by well-trained reviewers and all participating adolescent students gave written informed consent (or parents or legal guardians signed informed consent), participants used self-reported questionnaires in the school’s computer room to complete an online survey. Asthma was determined by physician diagnosis. Adolescents who did not provide sleep data (*n* = 235) and unresponsive data (*n* = 235) were excluded. Ultimately, 24,655 asthmatic adolescents and 253,775 non-asthmatic adolescents were included in the final analyses.

### 2.2. Sociodemographic Factors

The self-administered questionnaires explored socioeconomic factors: age, gender, type of school, family income, residential area, academic performance, health behavior factors (smoking, alcohol use, exercise, caffeine intake, sexual experience, and use of illegal drugs), and psychological factors (health status, stress level, and depression status). Family income was divided into high, middle, and low.

The residence area was categorized into urban and rural areas. Academic performance was assessed by asking the following question: “How do you assess your academic performance over the last year?” Smoking was indicated by a reply of “more than 1 day over the past month” to the question “Do you smoke?” [18]. Alcohol consumption was indicated by a reply of “more than 1 day over the past month” to the question ‘Do you drink alcohol?” [18]. Regular exercise status was explored by asking, “How many days of intense exercise causing an increased heart rate for at least 60 min have you performed over the last week?” Answers were classified as “no” (<3 times/week) or “yes” (≥3 times/week) [19]. Experience of sex was determined by the response to the question, “Have you ever had sexual intercourse with the same or different sex partner(s)?” Adolescents who ingest high doses of caffeine have reported sleep problems and morning fatigue [20]. Thus, we adjusted for caffeine intake as a potential confounding variable. Caffeine intake was classified into three categories: High (≥5 times/week), moderate (1–4 times/week), and infrequent (less than once/week) [21]. The experience of illicit drug use was also explored. Stress levels were classified as low (none to some) or high (moderate to very high). Health status was classified as healthy (fair to very good), fair, or bad (bad to very bad). To define depression, the International Diagnostic Interview—Short Form questionnaire [22] was employed; this has been found to be useful in large health surveys. This was completed by asking, “Did you have depressed feelings more than 2 weeks in the last year?”.

### 2.3. Sleep

The average sleep time for participant’s weekdays and weekends was calculated using responses to questions of the Korean version of the MCTQ. Average sleep duration was calculated by the following equation: (weekday sleep duration × 5 + weekend sleep duration × 2)/7. To determine chronotypes, we calculated the midpoint of sleep on school-free days (MSF), the midpoint of sleep on school days (MSW), sleep duration on school days (SDW), and sleep duration on school-free days (SDF). MSFsc (midpoint of sleep on school-free days corrected for sleep extension on school-free days) was used as the chronotype [23]. The MSFsc was determined using the following equation [24]: MSFsc = MSF − ((SDF − (SDW × 5 + SDF × 2)/7)/2. When SDF was shorter than or equal to SDW, MSFsc was the same as MSF. Chronotype was categorized as quintiles of the MSFsc: early chronotype (Q1, lowest MSFsc), intermediate chronotype (second (Q2), third (Q3), and fourth (Q4) quintiles), and late chronotype (Q5, highest MSFsc) [25]. Social jetlag was calculated from the following equation: social jetlag = MSF − MSW. Sleep satisfaction was classified into three groups: sufficient, moderate, and insufficient.

### 2.4. Data Analysis

The basic characteristics of the asthma and non-asthma groups were compared using the chi-squared test. Multiple logistic regression with complex sampling was performed. Complex sampling involves identifying and collecting data from population units chosen via multiple phases of identification and selection [26]. We adjusted for age, sex (Model 1) and additional socioeconomic factors (age, gender, type of school, family income, residential area, and academic performance), health behaviors (smoking, alcohol use, exercise, caffeine intake, sexual experience, and experience with illegal drugs) (Model 2) and psychological factors (health status, stress level, and depression status) (Model 3) when exploring associations between sleep chronotype and asthma. *p*-values less than 0.05 were taken to be statistically significant. SPSS for Windows was used for all data analyses (version 21.0; SPSS Inc., Chicago, IL, USA).

## 3. Results

Table 1 lists the subject demographic characteristics. Male sex, co-education and middle school type, living in an urban area, high and low family income, and high academic performance were significantly more common in asthmatics, as were smoking, alcohol intake, high and moderate caffeine intake, regular exercise, substance use, sexual experience, and psychosomatic factors such as poor health status, severe stress, unhappiness, and depressive mood.

Table 2 summarizes sleep parameters, which differed between the groups. The asthma group slept less than the non-asthma group (≤5 h: 24.3% vs. 23.2%; *p* < 0.001). Mean sleep duration (430.6 ± 95.6 vs. 433.5 ± 93.6 min; *p* < 0.001), MSF (255.9 ± 75.9 vs. 258.3 ± 73.6 min; *p* < 0.001), MSW (199.1 ± 49.1 vs. 200.1 ± 48.4 min; *p* < 0.001), SDW (398.2 ± 98.1 vs. 400.2 ± 96.8 min; *p* = 0.002), and SDF (511.8 ± 151.9 vs. 516.7 ± 147.2 min; *p* < 0.001) were significantly lower in the asthma group, as was social jetlag (56.8 ± 72.9 vs. 59.2 ± 70.9 min; *p* = 0.003). The late sleep chronotype was significantly more common in the asthma group than the non-asthma group (21.1% vs. 20.0%, *p* < 0.001). Poor sleep satisfaction was significantly more prevalent in the asthma than the non-asthma group (sleep satisfaction insufficient: 41.3% vs. 37.5%, *p* < 0.001). Table 3 shows the adjusted odds ratios (ORs) for adolescent asthma frequency by sleep chronotype. After adjusting for age and sex (Model 1), the OR of asthma frequency was 1.09-fold higher (95% confidence interval (CI) 1.05–1.12) in subjects with a late versus intermediate sleep chronotype. After adjusting for Model 1 factors plus sociodemographic factors (Model 2), late sleep chronotype was significantly associated with an increased frequency of asthma (OR 1.08; 95% CI 1.04–1.12) compared to the intermediate sleep chronotype. After adjusting for Model 2 factors plus psychosomatic factors (Model 3), late sleep chronotype was significantly associated with an increased risk of asthma (OR 1.05; 95% CI 1.01–1.09) compared to the intermediate sleep chronotype.

## 4. Discussion

Individual sleep/wake differences can be quantified using questionnaires such as the Horne–Östberg MEQ [27] and MCTQ [23] or via actigraphy [28].

Because the chronotypes form a normal distribution among adolescents, early or late chronotypes are uncommon and intermediate chronotypes are common [4]. Chronotypes depend on adolescent characteristics such as age or sex [3] and different environmental factors such as light exposure [29]. An understanding of chronobiology is important for medical professionals because chronotherapy can be used to treat asthma.

Late sleep chronotypes are associated with adolescent asthma. Asthma is characterized by chronic inflammatory airway disease, and inhaled or oral steroids are frequently used for asthma control. Deterioration of nocturnal asthma symptoms affecting the majority of adolescent asthma patients has long been considered a common symptom.

Although the association between late sleep chronotype and asthmatic activity remains unclear, it may be due to the abnormal increase in levels of bronchial eosinophils, and their mediators, such as histamine, leading to nocturnal airway hyperreactivity [30]. Furthermore, bronchial airway eosinophils are present in nocturnal asthma, suggesting that diurnal variation in the activation of airway inflammatory cells contributes to lung function fluctuations in asthma [31]. Circadian sleep behavior changes, such as sleep deprivation, are risk factors that can exacerbate inflammatory diseases such as asthma [32]. Many pulmonologists have shown that chronotherapy for asthma is a useful treatment, particularly for nocturnal asthma [33]. Ferraz et al. [34] explored whether asthma, with or without nocturnal exacerbations, showed a different chronotype distribution. Nocturnal asthma patients exhibited fewer morning chronotypes than other patients. However, the sleep chronotype distribution was not significantly different between asthmatic and non-asthmatic populations. In our study, asthma was associated with a late sleep chronotype. The association between late sleep chronotype and adolescent asthma suggests that the circadian clock may be associated with asthma pathogenesis. Because behavioral features of the early sleep chronotype/late sleep chronotype matches the circadian time duration, late sleep chronotypes have a longer circadian sleep time duration and early sleep chronotypes have shorter circadian sleep time durations. This does not suggest that early sleep chronotypes have a circadian pacemaking system dysfunction, but early sleep chronotypes tend to have an increased need for sleep to compensate for sleep needed during schooldays. However, late chronotypes tend to have circadian sleep disruption, which may develop into circadian sleep dysfunction.

Our study had some limitations. First, we used self-reported data, so the sleep time and wake time may be unreliable despite most adolescents correctly reporting their sleep time and wake time [35]. Also, asthma assessment was based only on physician diagnoses. Further prospective studies that employ more precise investigations of sleep time such as actigraphy are required to address this limitation. Second, our cross-sectional study design did not support a causal relationship between sleep chronotype and adolescent asthma. Furthermore, a cross-sectional study cannot adequately address cause and effect. Further studies, such as path analysis, are required to explore the association between sleep chronotype and adolescent asthma. Third, we did not investigate asthma-related information; the use of asthma medication or the dose of steroids may affect sleep, lung function data, or asthma control test scores, and we lacked information on current symptoms and treatments. Fourth, our study was based on web-based self-reported physician-diagnosed asthma and did not assess current asthma symptoms. Therefore, our results should be considered with caution regarding the association between asthma and sleep chronotype. Fifth, gastroesophageal reflux disease (GERD) [36], snoring [37], and oral steroid use might be associated with sleep disturbances, but we did not analyze these potential confounding factors. Finally, the observed association between late chronotype and asthma is very modest (OR of 1.05 in the fully adjusted model), so evidence for a causal association is weak; it is possible that the association could be explained by an unmeasured confounder, such as diet.

Despite the above limitations, this study had many strengths. As the KYRBWS data were gathered via nationwide weighted sampling, they are representative of all South Korean adolescents. For the treatment of asthma, sleep has received little attention. Many adolescent asthmatics suffer from sleep insufficiency, so increased attention to sleep issues is required for asthma control. However, further, well-designed, randomized prospective studies are required to explore the relationship between sleep chronotype and adolescent asthma.

## 5. Conclusions

In conclusion, although our study showed very modest association (OR of 1.05 in the fully adjusted model), we show that the late sleep chronotype is associated with asthma in adolescents in South Korea. Sleep and wake timing are controlled by individual circadian rhythms.

## Figures and Tables

**Table 1 ijerph-17-03000-t001:** General characteristics of participating adolescents.

	Asthma (*n* = 24,655)	No Asthma (*n* = 253,775)	*p*-Value
Sex			<0.001
Girl	10,691 (43.4)	128,807 (50.8)	
Boy	13,964 (56.6)	124,968 (49.2)	
Age	14.9 ± 1.7	15.0 ± 1.7	<0.001
BMI	21.2 ± 3.3	20.9 ± 3.2	<0.001
School			<0.001
Middle school	12,461 (50.5)	122,426 (48.2)	
Academic high school	10,129 (41.1)	108,167 (42.6)	
Vocational high school	2065 (8.4)	23,182 (9.1)	
School type			<0.001
Southern school	4955 (20.1)	44,488 (17.5)	
Girl school	3788 (15.4)	47,405 (18.7)	
Coeducation	15,912 (64.5)	161,882 (63.8)	
Living area			0.002
Rural	1628 (6.6)	18,130 (7.1)	
Urban	23,027 (93.4)	235,645 (92.9)	
Living			0.100
Living without parents	1183 (4.8)	11,594 (4.6)	
Living with parents	23,472 (95.2)	242,281 (95.4)	
Economic level			<0.001
Low	4463 (18.1)	42,685 (16.8)	
Medium	11,097 (45.0)	122,101 (48.1)	
High	9095 (36.9)	85,989 (35.1)	
Subjective academic achievement			<0.001
Low	8026 (32.6)	85,250 (33.6)	
Middle	6741 (27.3)	72,276 (28.5)	
High	9888 (40.1)	96,249 (37.9)	
Smoking	1060 (4.3)	9418 (3.7)	<0.001
Alcohol	1704 (6.9)	15,553 (6.1)	<0.001
Caffeine energy drink			<0.001
Infrequent (<1 time/week)	5052 (20.5)	50,543 (19.9)	
Moderate (1–4 times/week)	19,341 (78.4)	201,504 (79.4)	
Highly (≥5 times/week)	262 (1.1)	1728 (0.7)	
Exercise	8710 (35.3)	81,089 (32.0)	<0.001
Experience of sex	1059 (4.3)	8768 (3.5)	<0.001
Illegal substance use	236 (1.0)	1289 (0.5)	<0.001
Perceived Stress			<0.001
Severe	10,184 (41.3)	95,080 (37.5)	
Moderate	10,219 (41.4)	109,429 (43.1)	
None to mild	4252 (17.2)	49,266 (19.4)	
Perceived health status			<0.001
Healthy	22,266 (90.3)	239,020 (94.2)	
Bad	2389 (9.7)	14,755 (5.8)	
Perceived happiness			<0.001
Happy	22,054 (89.5)	121,589 (91.7)	
Unhappy	2601 (10.5)	21,186 (8.3)	
Depressive mood	7493 (30.4)	64,528 (25.4)	<0.001

Data are presented as *n* (percent) or mean ± SD.

**Table 2 ijerph-17-03000-t002:** Sleep variables in asthma and non-asthma adolescents.

	Asthma (*n* = 24,655)	No asthma (*n* = 253,775)	*p*-Value
Sleep time			<0.001
≤5 h	5992 (24.3)	58,909 (23.2)	
6 h	5398 (21.9)	54,819 (21.6)	
7 h	5390 (21.9)	56,331 (22.2)	
8 h	4698 (19.1)	50,974 (20.1)	
≥9 h	3177 (12.9)	32,742 (12.9)	
Sleep duration, average, min	430.6 ± 95.6	433.5 ± 93.6	<0.001
Midpoint of sleep on school free days (MSF)	255.9 ± 75.9	258.3 ± 73.6	<0.001
Midpoint of sleep on school days (MSW)	199.1 ± 49.1	200.1 ± 48.4	<0.001
Sleep duration on school days (SDW)	398.2 ± 98.1	400.2 ± 96.8	0.002
Sleep duration on school free days (SDF)	511.8 ± 151.9	516.7 ± 147.2	<0.001
Chronotype			<0.001
Q1: early	4946 (20.1)	50,740 (20.0)	
Q2: intermediate	4962 (20.1)	50,724 (20.0)	
Q3: intermediate	4812 (19.5)	50,870 (20.0)	
Q4: intermediate	4728 (19.2)	50,842 (20.0)	
Q5: late	5207 (21.1)	50,599 (20.0)	
Social jetlag, min	56.8 ± 72.9	59.2 ± 70.9	0.003
Sleep satisfaction			<0.001
Enough	4252 (17.2)	49,296 (19.4)	
Moderate	10,219 (41.4)	109,429 (43.1)	
Not enough	10,184 (41.3)	95,080 (37.5)	

Data are shown as N (percent) or mean ± SD.

**Table 3 ijerph-17-03000-t003:** Adjusted odds ratio for adolescent asthma risk by sleep chronotype.

Sleep Duration	Model 1 OR (95% CI)	Model 2 OR (95% CI)	Model 3 OR (95% CI)
Q2–4: intermediate	reference	reference	reference
Q1: early	1.00 (0.97–1.04)	0.99 (0.96–1.03)	1.01 (0.97–1.04)
Q5: late	1.09 (1.05–1.12)	1.08 (1.04–1.12)	1.05 (1.01–1.09)

Data are presented as odds ratios (OR) and 95% confidence intervals (CI). Model 1: adjusted for age and sex. Model 2: adjusted for age, sex, BMI, smoking, alcohol, exercise, family income, living area, school type, experience of sex, illegal drug use, academic performance, caffeine intake, and living with parents. Model 3: adjusted for age, sex, BMI, smoking, alcohol, exercise, family income, living area, school type, experience of sex, illegal drug use, academic performance, caffeine intake, living with parents, perceived stress, perceived health status, perceived happiness, and depressive mood.

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
