# Peer review of "Late Chronotype is Associated with Adolescent Asthma: Assessment Using the Korean-Version MCTQ"

_ijerph, 2020, doi:10.3390/ijerph17093000_

Round 1

Reviewer 1 Report

This was an interesting manuscript that investigated the relationship between chronotype and asthma in a large adolescent sample. Very small relationships were also found between asthma and sleep variables.

While this was a very interesting manuscript, I have a few concerns:

  • It is my opinion that the introduction needs to be restructured. It doesn't touch on the importance of sleep for adolescents, nor the adolescent change to greater evening preference. The link between asthma and chronotype is unclear, as is the mention of smoking behaviour. I wondered whether smoking behaviour was discussed because of the relationship between risk behaviours and later chronotype preference? 
  • In the results section, you have said that living with parents is significantly associated with asthma, but Table 1 says p=0.100 for this relationship
  • I think the discussion would also benefit from a restructure. It's not clear to me why there is a discussion of the MEQ versus the MCTQ - I feel that justification of the measures used should be brief and in the methods section rather than so long and in the discussion. I also feel that the discussion of the importance of measuring chronotype and the use of chronotherapy should be moved to the introduction as this is the justification for the study. 

Author Response

This was an interesting manuscript that investigated the relationship between chronotype and asthma in a large adolescent sample. Very small relationships were also found between asthma and sleep variables.

While this was a very interesting manuscript, I have a few concerns:

  • It is my opinion that the introduction needs to be restructured. It doesn't touch on the importance of sleep for adolescents, nor the adolescent change to greater evening preference. The link between asthma and chronotype is unclear, as is the mention of smoking behaviour. I wondered whether smoking behaviour was discussed because of the relationship between risk behaviours and later chronotype preference? 

Answer) I corrected as your recommendation introduction part more clearly as follows

As you wonder about association between late chronotype and smoking (Addiction , 106 (1), 170-7 Jan 2011 Evening Types Are More Often Current Smokers and Nicotine-Dependent-A Study of Finnish Adult Twins) showed that evening types were much more likely to be current (OR = 2.91, 95% CI 2.50, 3.38) and life-time smokers (OR = 2.67, 95% CI 2.96, 4.07) compared to morning types. The association between diurnal type and smoking or nicotine dependence could be partly explained by nicotine's pharmacological properties. Nicotine is a stimulant and this could make smokers more alert and stay up later. If this were the case, smoking cessation would result in a lower tendency to be an evening type. However, our study assessed diurnal type only once, so we cannot examine this possibility.

  • In the results section, you have said that living with parents is significantly associated with asthma, but Table 1 says p=0.100 for this relationship

Answer) I deleted living with parents as your recommendation

  • I think the discussion would also benefit from a restructure. It's not clear to me why there is a discussion of the MEQ versus the MCTQ - I feel that justification of the measures used should be brief and in the methods section rather than so long and in the discussion. I also feel that the discussion of the importance of measuring chronotype and the use of chronotherapy should be moved to the introduction as this is the justification for the study. 

Answer) I corrected as your recommendation

Reviewer 2 Report

This study examines the association between sleep parameters (duration, chronotypes) and asthma in a large cohort of Korean adolescents. The authors should clarify their hypotheses for investigating this question. In addition, they should modulate their conclusions regarding the association between late chronotype and asthma as the observed association is very modest (adjusted OR: 1.05), and might thus be explained by unmeasured confounding.

Major comments:

Introduction

  1. The authors’ hypotheses should be clarified. First, it seems from the introduction that asthma can influence sleep duration and quality, but they also hypothesize that sleep could influence asthma (development? symptoms?). Also, they seem to assume that sleep chronotype can impact asthma (“few studies have explored the effect of sleep chronotype on asthma”). The authors should clarify the rationale for investigating this association, and refrain to use words “influence” or “effect” unless it is clearly justified.
  2. P2, l 56-57 What are the results on the association between chronotypes and asthma in other populations? (adults? younger children?)

Methods

  1. Data from the Korea Youth Risk Behavior 66 Web-based Survey (KYRBWS) are used. Please provide details of recruitment, participation rate, missing data (flow chart) for this cohort and selection of the study population. Please also provide key references, if available.
  2. P2, l 90: please clarify what it means “to adjust for caffeine intake when defining sleep chronotype”? It seems that caffeine was adjusted for in multivariable models, as the other confounders.
  3. P3 l 115: Please clarify what is the “complex sampling analysis” and provide reference.

Results

  1. Table 1: p is missing for several factors: alcohol, exercise, experience of sex, illegal substances.
  2. P3 l 125: “living with/without parents” is incorrectly listed as a factor significantly associated with asthma (p=0.10 in Table 1).
  3. Table 1. For income, both high and low income associated with asthma. This is similar for caffeine intake. This should be described accordingly in text.
  4. P4 l 141: the OR of 1.19 is incorrectly reported (should be 1.09)

Discussion

  1. My major comment is that the authors should modulate their conclusions as observed associations are very modest (1.05 in fully adjusted model for late chronotype), even they are statistically significant (very large population). The hypotheses to support a causal association are thin, and there is a high probability that this association could be explained by unmeasured confounding. The authors should make this clear, for instance by calculating the E-Value (see . VanderWeele et al., Sensitivity Analysis in Observational Research: Introducing the E-Value, Ann Intern Med 2017. doi:10.7326/M16-2607). One confounder not accounted for, for instance, is diet and could probably by itself explain the observed association.
  2. Several additional limitations should be stated/clarified: first, asthma assessment is limited, as it is only based on physician-diagnosed asthma evaluated by questionnaire, and no information is available on current symptoms and treatment. Moreover, this cross-sectional study cannot inform on the direction of the association. This should be stated more clearly.
  3. P 6, L 187-190. The interest of chronotherapy for the treatment of asthma is largely beyond the scope of this paper (especially given the very small observed effect size), and does not seem well supported by the literature (the cited reference (35,36) do not address asthma specifically).
  4. P 6, l 196-199: references need to be added to support this assumption.

  1. The manuscript should be entirely checked for potential reporting errors (beyond the ones I have noticed) and English language editing is also needed. One author seems to be missing in the manuscript.

Author Response

This study examines the association between sleep parameters (duration, chronotypes) and asthma in a large cohort of Korean adolescents. The authors should clarify their hypotheses for investigating this question. In addition, they should modulate their conclusions regarding the association between late chronotype and asthma as the observed association is very modest (adjusted OR: 1.05), and might thus be explained by unmeasured confounding.

Major comments:

Introduction

  1. The authors’ hypotheses should be clarified. First, it seems from the introduction that asthma can influence sleep duration and quality, but they also hypothesize that sleep could influence asthma (development? symptoms?). Also, they seem to assume that sleep chronotype can impact asthma (“few studies have explored the effect of sleep chronotype on asthma”). The authors should clarify the rationale for investigating this association, and refrain to use words “influence” or “effect” unless it is clearly justified.

Answer) I corrected as your recommendation as follows

Asthma symptoms are often exacerbated at night through physiobiological changes (e.g.,airway inflammation, accidental cough, wheezing, and breathing difficulty). Asthma might have an adverse effect on sleep biology, but few reports have investigated whether sleep chronotype might affect asthma. Characterizing different chronotypes may increase our understanding of the relationship between disease and circadian rhythm. Chronotype is affected by several factors such as age, sex, residence area, geographical location, and job types. Although different chronotypes have been investigated in the general adult population, but studies regarding on adolescent asthma are not investigated. Thus, we explored the sleep chronotype characteristics of asthmatic and non-asthmatic adolescent patients and analyzed whether there is an association between sleep chronotype and adolescent asthma based on nationally representative school-based Korean adolescent self-reported data.

  1. P2, l 56-57 What are the results on the association between chronotypes and asthma in other populations? (adults? younger children?)

 Answer) I corrected as your recommendation as follows

Although different chronotypes have been investigated in the general adult population

Methods

  1. Data from the Korea Youth Risk Behavior 66 Web-based Survey (KYRBWS) are used. Please provide details of recruitment, participation rate, missing data (flow chart) for this cohort and selection of the study population. Please also provide key references, if available.

 Answer) I corrected as your recommendation as follows

The adolescents were stratified by 43 regional and school-type variables, and samples were then selected by reference to school and class. The survey, which was conducted online, was an anonymous self-administered questionnaire comprising 134 questions on 14 categories of health behaviors in Korean adolescents and showing a participation rate of 95.5% (17).

  1. P2, l 90: please clarify what it means “to adjust for caffeine intake when defining sleep chronotype”? It seems that caffeine was adjusted for in multivariable models, as the other confounders.

Answer)  I corrected more clearly as your recommendation as follows

Thus, we adjusted caffeine intake for confounding variable

  1. P3 l 115: Please clarify what is the “complex sampling analysis” and provide reference.

Answer)  I corrected more clearly and add reference as your recommendation as follows

Multiple logistic regression with complex sampling analysis (complex sample surveys involve the identification and data collection of a sample of population units via multiple stages or phases of identification and selection (27))

Results

  1. Table 1: p is missing for several factors: alcohol, exercise, experience of sex, illegal substances.

Answer) I corrected as your recommendation

Alcohol

1704 (6.9)

15553 (6.1)

<0.001

Caffeine energy drink

<0.001

 Infrequent (<1 time/week)

5052 (20.5)

50543 (19.9)

 Moderate (1-4 times/week)

19341 (78.4)

201504 (79.4)

 Highly (≥5 times/week )

262 (1.1)

1728 (0.7)

Exercise

8710 (35.3)

81089 (32.0)

<0.001

Experience of sex

1059 (4.3)

8768 (3.5)

<0.001

Illegal substance use

236 (1.0)

1289 (0.5)

<0.001

  1. P3 l 125: “living with/without parents” is incorrectly listed as a factor significantly associated with asthma (p=0.10 in Table 1).

Answer) I corrected as your recommendation as follows

Male sex, co-education and middle school type, living in urban area, high and low economic level of income, and high academic performance were significantly higher in asthmatics, as were smoking, alcohol intake, high and moderate caffeine intake, regular exercise, substance use, and sexual experience as well as pyschosomatic factors such as poor health status, severe stress, unhappiness feeling, and more depression mood.

  1. Table 1. For income, both high and low income associated with asthma. This is similar for caffeine intake. This should be described accordingly in text.

Answer) I corrected as your recommendation as follows

Male sex, co-education and middle school type, living in urban area, high and low economic level of income, and high academic performance were significantly higher in asthmatics, as were smoking, alcohol intake, high and moderate caffeine intake, regular exercise, substance use, and sexual experience as well as pyschosomatic factors such as poor health status, severe stress, unhappiness feeling, and more depression mood.

  1. P4 l 141: the OR of 1.19 is incorrectly reported (should be 1.09)

Answer) I corrected as your recommendation

Discussion

  1. My major comment is that the authors should modulate their conclusions as observed associations are very modest (1.05 in fully adjusted model for late chronotype), even they are statistically significant (very large population). The hypotheses to support a causal association are thin, and there is a high probability that this association could be explained by unmeasured confounding. The authors should make this clear, for instance by calculating the E-Value (see . VanderWeele et al., Sensitivity Analysis in Observational Research: Introducing the E-Value, Ann Intern Med 2017. doi:10.7326/M16-2607). One confounder not accounted for, for instance, is diet and could probably by itself explain the observed association.

Answer) Thank you for your sharp comments. I inserted this limitation as your recommendation as follows. As your recommendation E-value can be more clear tools for our marginal result, but unfortunately our SPSS statistics cannot calculate E-value.

Finally, our study observed associations are very modest (OR for 1.05 in fully adjusted model for late chronotype), even they are statistically significant. The hypotheses to support a causal association are thin, and there is a high probability that this association could be explained by unmeasured confounding such as diet.

  1. Several additional limitations should be stated/clarified: first, asthma assessment is limited, as it is only based on physician-diagnosed asthma evaluated by questionnaire, and no information is available on current symptoms and treatment. Moreover, this cross-sectional study cannot inform on the direction of the association. This should be stated more clearly.

Answer) Thank you for your sharp comments. I inserted this limitation as your recommendation as follows

In this study had some limitations. First, we used self-reported data, so the sleep time and wake time may be unreliable despite most adolescents correctly reporting their sleep time and wake time (40) and asthma assessment is limited, as it is only based on physician-diagnosed asthma evaluated by questionnaire. Further prospective studies that employ more precise investigations of sleep time such as actigraphy are required to solve this limitation. Second, our cross-sectional study design did not support the causal relationship between sleep chronotype and adolescent asthma. Furthermore, this cross-section design cannot inform on the direction of the association. Further studies are required to explore the associations between sleep chronotype and adolescent asthma. Third, we did not investigate asthma-related information; the use of asthma medication or the dose of steroids may affect sleep, lung function datas, or asthma control test scores and no information is available on current symptoms and treatment. Fourth, our study was based on web-based self-reported physician diagnosed asthma and did not assess current asthma symptoms. Therefore, our results should be considered with caution regarding the association between asthma and sleep chronotype. Fifth, gastroesophageal reflux diseases (GERD) (41), snoring (42), and oral steroid might have association with sleep disturbances, but we did not explore these confounding factors. Finally, our study observed associations are very modest (OR for 1.05 in fully adjusted model for late chronotype), even they are statistically significant. The hypotheses to support a causal association are thin, and there is a high probability that this association could be explained by unmeasured confounding such as diet.

  1. P 6, L 187-190. The interest of chronotherapy for the treatment of asthma is largely beyond the scope of this paper (especially given the very small observed effect size), and does not seem well supported by the literature (the cited reference (35,36) do not address asthma specifically).

Answer) I corrected as your recommendation so delete this phrase

  1. P 6, l 196-199: references need to be added to support this assumption.

Answer) I inserted references as your recommendation

  1. The manuscript should be entirely checked for potential reporting errors (beyond the ones I have noticed) and English language editing is also needed. One author seems to be missing in the manuscript.

Answer) I corrected as your recommendation

We checked English editing and following phrase as proof

The English in this document has been checked by at least two professional editors, both native speakers of English. For a certificate, please see:

http://www.textcheck.com/certificate/3BmZe4

(1) We recommend that you include the above statement in the end of your document to inform reviewers that the English has been professionally checked. If you do not, then you may receive the comment "English needs revision". Some reviewers who are not native speakers of English seem to add the comment "English needs revision" to EVERY paper they review by non-US authors... (The above text can be deleted later).

(2) We strongly recommend that you do not make ANY changes to this document. Textcheck should be the last step before final formatting. No further editing should be necessary unless there is text that the editors have misunderstood.

If any text has been misunderstood:

Rewrite the sentence(s) that you need to change, SAVE, and upload the completed document to:

http://www.textcheck.com/client/submit

for a final check.  (Please do not mark your changes in any way and do not send the document to us by e-mail unless you have a problem with the www site.)

Requests for a final check should be made within 1 month.

[If you make changes to this document and require a re-edit (such as changes post review) then it should be uploaded via 'Submit Document' on our www site, with a note that the file is a revision of '20041023'. The fee for a revised document is based on the wordcount of changed and new sentences (only).  As there is no charge for sentences that have not changed, we recommend that clients always upload complete documents.]

Round 2

Reviewer 2 Report

Most of my comments were addressed except the first one regarding the introduction:

I did not see any clarification of the hypothesis in the author’s response. Also, just replacing the word “influence” by “affects” (which is basically a synonym) does not solve the problem. This cross-sectional study cannot investigate whether sleep affects/influence asthma, but only the association between sleep parameters and asthma. I believe that the underlying hypothesis is that the direction of the association could be both ways: asthma can influence sleep and sleep can influence asthma, but this is still unclear.

Author Response

Answer) Thank you for your sharp comments. As you pointed, our cross-section design n a cross-sectional study, data are collected on the whole study population at a single point in time to examine the relationship between disease (or other health related state) and other variables of interest. Cross-sectional studies therefore provide a snapshot of the frequency of a disease or other health related characteristics in a population at a given point in time. This methodology can be used to assess the burden of disease or health needs of a population, for example, and is therefore particularly useful in informing the planning and allocation of health resources. So difficult to determine whether the outcome followed exposure in time or exposure resulted from the outcome. As you comments, asthma can influence sleep and sleep can influence asthma. Previous study investigated sleep problem in asthma, but no report on sleep can affect asthma. As you commented our study not guaranteed late sleep chronotype might cause asthma incidence, this problem can be partly solved path analysis. We corrected as your recommendation in discusstion as follows.

Further studies such as path analysis are required to explore the association between sleep chronotype and adolescent asthma
